# Chemical unfolding of protein domains induces shape change in programmed protein hydrogels

Luai R. Khoury[1]* & Ionel Popa [1]*

Programmable behavior combined with tailored stiffness and tunable biomechanical response are key requirements for developing successful materials. However, these properties are still an elusive goal for protein-based biomaterials. Here, we use protein-polymer interactions to manipulate the stiffness of protein-based hydrogels made from bovine serum albumin (BSA) by using polyelectrolytes such as polyethyleneimine (PEI) and poly-L-lysine (PLL) at various concentrations. This approach confers protein-hydrogels with tunable wide-range stiffness, from ~10–64 kPa, without affecting the protein mechanics and nanostructure. We use the 6-fold increase in stiffness induced by PEI to program BSA hydrogels in various shapes. By utilizing the characteristic protein unfolding we can induce reversible shape-memory behavior of these composite materials using chemical denaturing solutions. The approach demonstrated here, based on protein engineering and polymer reinforcing, may enable the development and investigation of smart biomaterials and extend protein hydrogel capabilities beyond their conventional applications.

---

[1] Department of Physics, University of Wisconsin-Milwaukee, 3135 North Maryland Ave., Milwaukee, WI 53211, USA. *email: khoury@uwm.edu; popa@uwm.edu

Polymer-based hydrogels have found broad applications in tissue engineering, drug delivery, soft robotics, and actuators[1,2], and their viscoelasticity can drive stem cell fate and activity[3]. However, these hydrogels possess limited mechanical strength, are prone to permanent breakage, lack dynamic switches, and reversible shape[2]. Several approaches have been proposed to improve their stiffness and extensibility. One method is through using a double-network crosslinking strategy, either by secondary polymer network, using multivalent ions, or by nanoparticles[4–10]. Furthermore, polymer hydrogels are being used in the shape-memory field. For example, thermoplastic polymers-based hydrogels can display shape-memory response as a function of temperature, which is of great importance to soft robotics and biomedical applications[11]. Additionally, supramolecular interactions between chains based on reversible hydrogen bonds, metal-coordination, or dynamic covalent bonds have been recently introduced to improve hydrogels functionality, but in all these examples the structure of the primary network changes during shape-morphing cycles[8,12,13].

In the last decade, protein hydrogels based on globular proteins were proposed as a novel biomaterial that may have a wide use in biomedical applications and research[14]. These hydrogels are intrinsically biocompatible, biologically diverse, and can use the unfolding response or tertiary structure for energy storage and release[10,15–17]. Currently the stiffness of protein-based hydrogels has a narrow tunability range, limited at the lower end by the minimum protein concentration required for gelation, and at the higher end by the solubility of the protein[10,16]. It has been challenging to obtain the same smart behavior as that of polymer-based hydrogels, in part due to the limited range of solvents, temperatures, and concentrations that can be used. Proteins generally require water-based solvents, a narrow range of salt concentrations and pH, and the working temperature to obtain biomaterials cannot exceed values well above 37 °C. Furthermore, the range of concentrations that can be utilized to obtain hydrogels is narrow[14,16,18]. At the lower end, a too low protein concentration leads to incomplete network formation. This incomplete crosslinking results in soft gels, showing irreversible deformations under strain. At the upper end, while the final stiffness of protein hydrogels can be improved with increasing protein concentration, and hence the crosslinking density, a major limitation comes from the maximum protein solubility. For example, hydrogels made from protein G, domain B1 (GB1), from SH3 or from chimera GB1-HP67 had a minimum gelation concentration of ~150 mg/mL and reached their solubility limit at ~180 mg/mL (which corresponds to ~1.3–3.2 mM)[18]. In this concentration range, the gels have a narrow change in stiffness of ~18%. For bovine serum albumin (BSA)-based hydrogels, concentrations below 1 mM (~6 kPa) produce gels showing plastic deformation under force, while the maximum solubility of BSA (~4 mM) only yields hydrogels with stiffness of ~15 kPa, setting the upper limit achievable with this method[15]. An increase in the stiffness range for protein hydrogels would not only expand their applications but would also allow for shape programmable behavior. Such a shape-memory approach based on protein (un)folding transitions does not currently exist.

Here, we report a method of producing hybrid protein-polymer hydrogels which have covalently crosslinked protein network reinforced with physically adsorbed polyelectrolytes. We use a custom-made force rheometer which utilizes an analog feed-back to expose protein-based hydrogels to various force protocols. We characterize the intake of various polyelectrolytes and determine the change in stiffness and folding of BSA-based hydrogels. We find that in the presence of branched-polyethyleneimine, BSA-based hydrogel can stiffen up to sixfold without affecting the unfolding nanomechanics of proteins

domains. Using this interaction between BSA and polymers, in combination with the unfolding response of protein domains in chemical denaturants, we formulate protein-based hydrogels to display reversible shape-memory behavior.

## Results

**Polyelectrolytes can stiffen protein-based hydrogels.** BSA is one of the most inexpensive and abundant proteins available. It has an overall negative charge at pH ~7.4 due to several negatively charged amino acids patches distributed on its surface[19]. Our first goal here was to determine the appropriate polyelectrolytes that can adsorb on BSA domains inside the hydrogel matrix. First, hydrogels were synthesized inside semi-transparent tubes (made from PTFE with inner diameter 558.8 μm) using 2 mM BSA and a photo-activated crosslinking reaction[15,20–22]. BSA has eight exposed tyrosine amino acids that can participate in the cross-linking reaction[23]. These covalently crosslinked hydrogels form a stable primary network, as at the chosen concentration of 2 mM, all protein domains are crosslinked[15]. Following gelation, the BSA hydrogels were equilibrated in TRIS buffer (Tris 20 mM, NaCl 150 mM, pH ~7.4) for 30 min at room temperature, then moved to one of the following polymer solutions: branched-polyethyleneimine (PEI) 10 kDa, poly-(L)-lysine (PLL) 10 kDa, and polyethylene glycol (PEG) 8 kDa, which were dissolved in the same TRIS buffer (Fig. 1a). Following incubation for another 30 min in one of a specific polymer concentration at room temperature, the BSA hydrogels were moved back in TRIS buffer, to wash any unbounded polymer molecules from the treated samples. The hydrogels were then characterized using force-clamp rheometry[15], scanning electron microscopy (SEM), and water content measurements.

The mechanical response of both native and polymer-treated hydrogels was measured using a force-ramp protocol where the stress was linearly increased and decreased with a rate of 40 Pa/s. Using this approach, we determine the Young's modulus from the initial slope of each stress–strain curve, which reports on the gel stiffness. Furthermore, as proteins unfold and refold at vastly different forces[24], the stress–strain curves show important hysteresis, which reports on the energy being dissipated during stress–relaxation cycles[20,25].

When the hydrogels are treated with a constant 1 mM polymer concentration, PEI had the largest effect on the measured stiffness, which increased from 10 ± 2 to 64 ±7 kPa. PLL increased the Young's modulus to 16 ± 1 kPa, while PEG did not induce any change (Fig. 1b, c). This trend was mirrored by the hydrogel structure when characterized by SEM, where no significant change in pore size was seen upon incubation with PEG (pore size 1126 ± 636 μm² for native-BSA, 1121 ± 700 μm² for PEG). PLL and PEI induced a decrease in the pore size, with areas of 502 ± 24 and 421 ± 141 μm², respectively (Fig. 1d, e, Supplementary Fig. 1).

Our rheometry-based approach allows us to easily asses the change in stiffness of BSA hydrogels treated with different concentrations of polyelectrolytes (Fig. 2). PEG-treated hydrogels, where the polymer was in 0.25–3 mM range, did not show any increase in the Young's modulus (Fig. 2a (iii)). On the other hand, both PLL and PEI resulted in the stiffening of the exposed BSA hydrogel (Fig. 2a (i) and Fig. 2a (ii)). The effect is more pronounced with PEI, where we measure up to sixfold increase in the gel stiffness (Fig. 2b). While both polymers are positively charged at pH 7.4 and with similar molecular weight, PEI is branched and has a higher charge density, and hence interacts more efficiently with the negatively charged amino acid patches on the BSA surface[26]. The increase in stiffness is mirrored also by the decrease in the hysteresis measured from stress–curve for

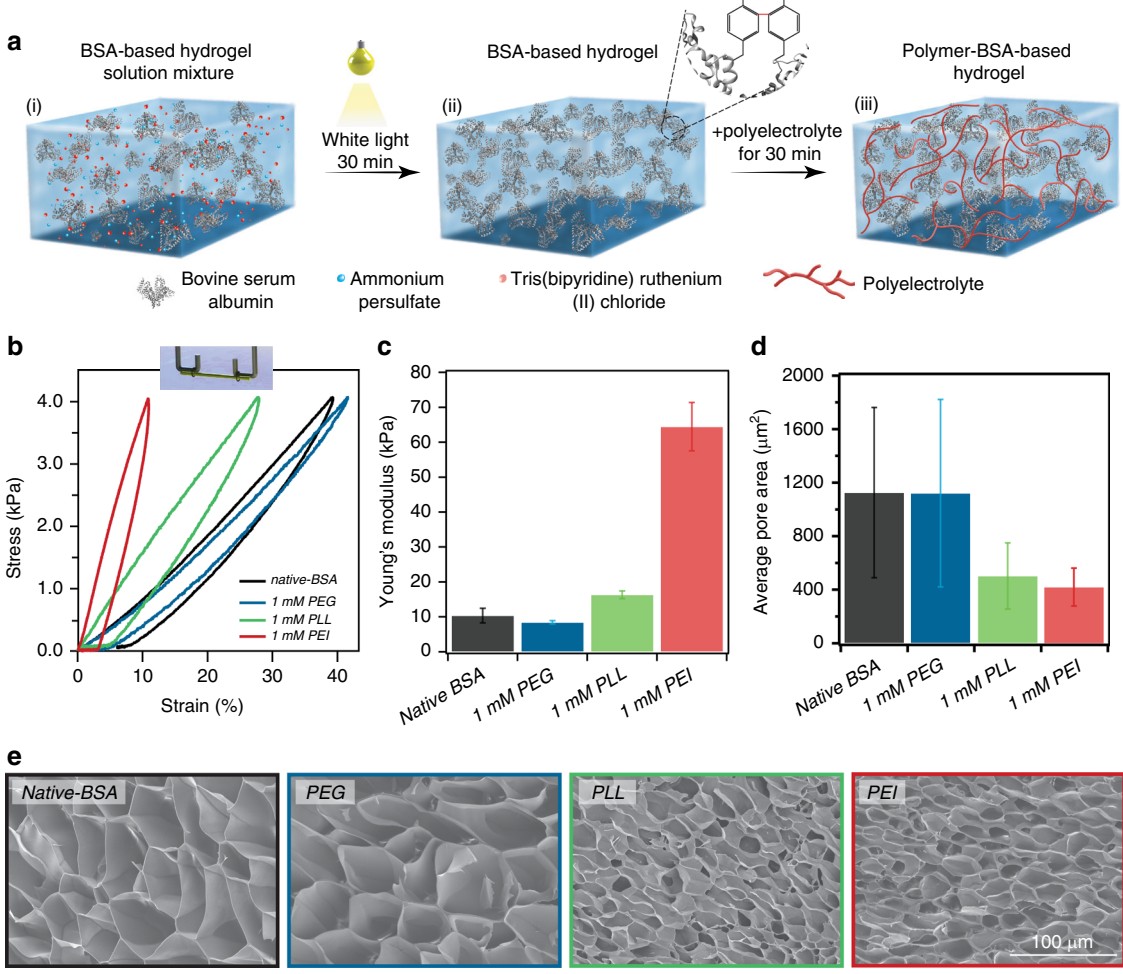

**Fig. 1** Synthesis of protein-based hydrogels treated with polyelectrolytes. **a** Strengthening process of BSA-based hydrogel using polymer–protein interaction: (i) BSA, ammonium persulfate (APS), and Tris(bipyridine) ruthenium (II) chloride ([(Ru(bpy)$_3$]$^{2+}$) are mixed together; (ii) the hydrogel mixture is exposed to white light for 30 min at room temperature (RT), which leads to covalent crosslinking between BSA domains via adjacent exposed tyrosines amino acids (inset); afterwards, the hydrogel is extruded into the TRIS solution; (iii) the hydrogel is treated with one of three polymer solutions for 30 min: PEI, PLL, or PEG, all dissolved in TRIS buffer. Thereafter, the hydrogel is moved back to TRIS solution, to remove any unbounded polymer molecules. **b** Stress–strain curves of native-BSA (black) and after incubation with 1 mM PEG (blue), PLL (green), and PEI (red). Inset: Scheme of the tethered hydrogels to the force-clamp (FC) rheometer hooks. **c** Average Young's moduli calculated from stress–strain curves of *native*-BSA, and when treated 1 mM of PEG (blue), PLL (green), and PEI (red). **d** Average pore-size values of native-BSA (black), and after incubation with 1 mM PEG (blue), PLL (green), and PEI (red) samples, as derived from SEM images analysis. **e** SEM images of native-BSA, and after incubation with the same polyelectrolytes. All error bars are SD from n = 3 independent hydrogel samples.

each polymer concentration (Fig. 2c). Additionally, measurement of the pore-size for each treated hydrogel sample using SEM showed that there is a shrinking trend in the pore size from 1126 ± 636 to 359 ± 216 μm$^2$ with increasing PEI concentration, which correlates with the slight decrease in water content (Fig. 2d–f). Interestingly, the wall thickness of the pores showed a positive correlation when increasing the PEI concertation up to 0.75 mM (Fig. 2g, Supplementary Fig. 2). This increase in wall thickness is probably due to an increase in polymer mass inside the hydrogel network, resulting in enhanced crosslinking[5]. Furthermore, treating BSA hydrogels with various amounts of PEI does not affect the folding of BSA domains inside the gel (Supplementary Fig. 3).

**Stiffening protein hydrogels with polyelectrolytes**. To decouple the response of protein (un)folding mechanics from the intrinsic elasticity coming from the polymer–protein interaction, we used guanidinium hydrochloride (GuHCl) 6 M, which acts as a

chemical denaturant[20]. Addition of the GuHCl to native-BSA hydrogels softens the gels and removes the hysteresis[15,27]. The softening comes from the fact that folded linked proteins are ~20× stiffer than unfolded polypeptide chains[28]. The disappearance of the hysteresis in stress–strain curves is a benchmark for the lack of tertiary and secondary structure, here induced through chemical denaturation (Fig. 3a, b). This interpretation is further supported from the decrease in fluorescence intensity of 8-Anilinonaphthalene-1-sulfonic acid (ANS)[29,30], which reports on the folding of BSA domains (Supplementary Fig. 3). Interestingly, when adding GuHCl to protein hydrogels treated with PEI, the hysteresis disappears as expected, but the stiffness is higher than that of the native-BSA gels in TRIS (~18 vs ~10 kPa; Fig. 3a, b), suggesting that the interaction between BSA and PEI remains strong and limits gel extensibility even in harsh conditions. After washing out the GuHCl salts from the PEI-treated hydrogel sample by immersing it in TRIS solution, the BSA domains refold back to their native state and the hydrogel regains its initial stiffness (~64 kPa) and shows a similar

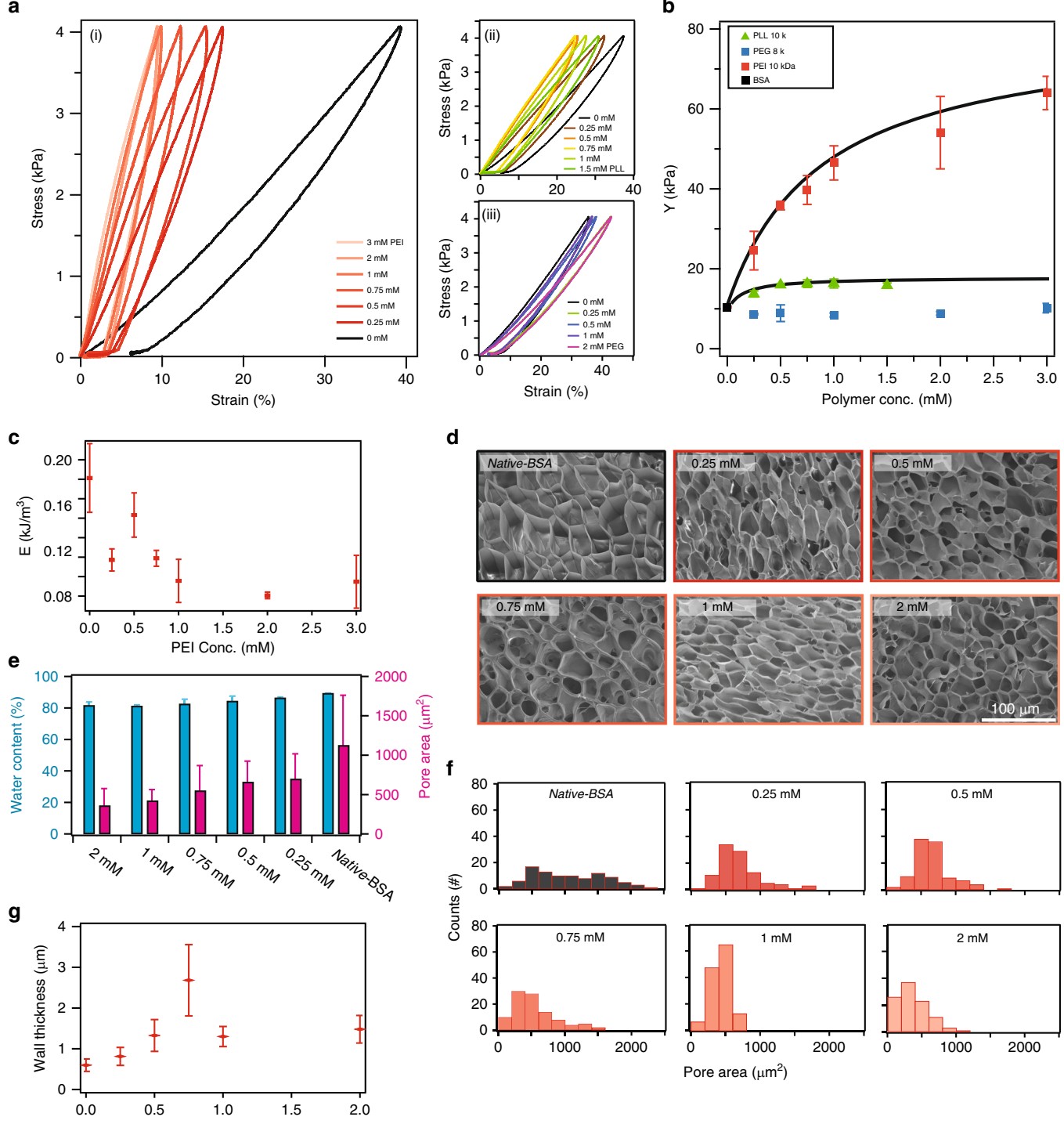

**Fig. 2** Characterizations of BSA hydrogels as a function of polyelectrolyte intake. **a** (i) Stress–strain curves of *native*-BSA, and incubated with various concentrations of PEI ranging from 0 to 3 mM; (ii) stress–strain curves of *native*-BSA and incubated with various concentrations of PLL ranging from 0 to 1.5 mM; (iii) stress–strain curves of different BSA-based hydrogels treated with various PEG concentrations ranging from 0 to 3 mM. Each curve represents the average of three different measurements. **b** Average Young's moduli calculated from stress–strain curves of *native*-BSA, and incubated with PEG (blue), PLL (green), and PEI (red) as a function of the polymer concentrations. Fits represent a Langmuir-like behavior, with equilibrium constants of $K_{PLL} = 1.1$ and $K_{PEI} = 5.3$ mM$^{-1}$. **c** Relationship between PEI concentration and energy dissipation of treated BSA hydrogels. The energy dissipation was calculated from the hysteresis area enclosed in the stress–strain curves. **d** SEM images of BSA-based hydrogels incubated with PEI at various concentrations. **e** Average pore-area size and water content measurements of native and PEI-treated hydrogel samples. **f** Pore-area size distribution histograms as derived from SEM images. **g** The relationship between wall thickness of the pores and PEI concentration. Error bars in all panels are SD from $n = 3$ independent hydrogel samples.

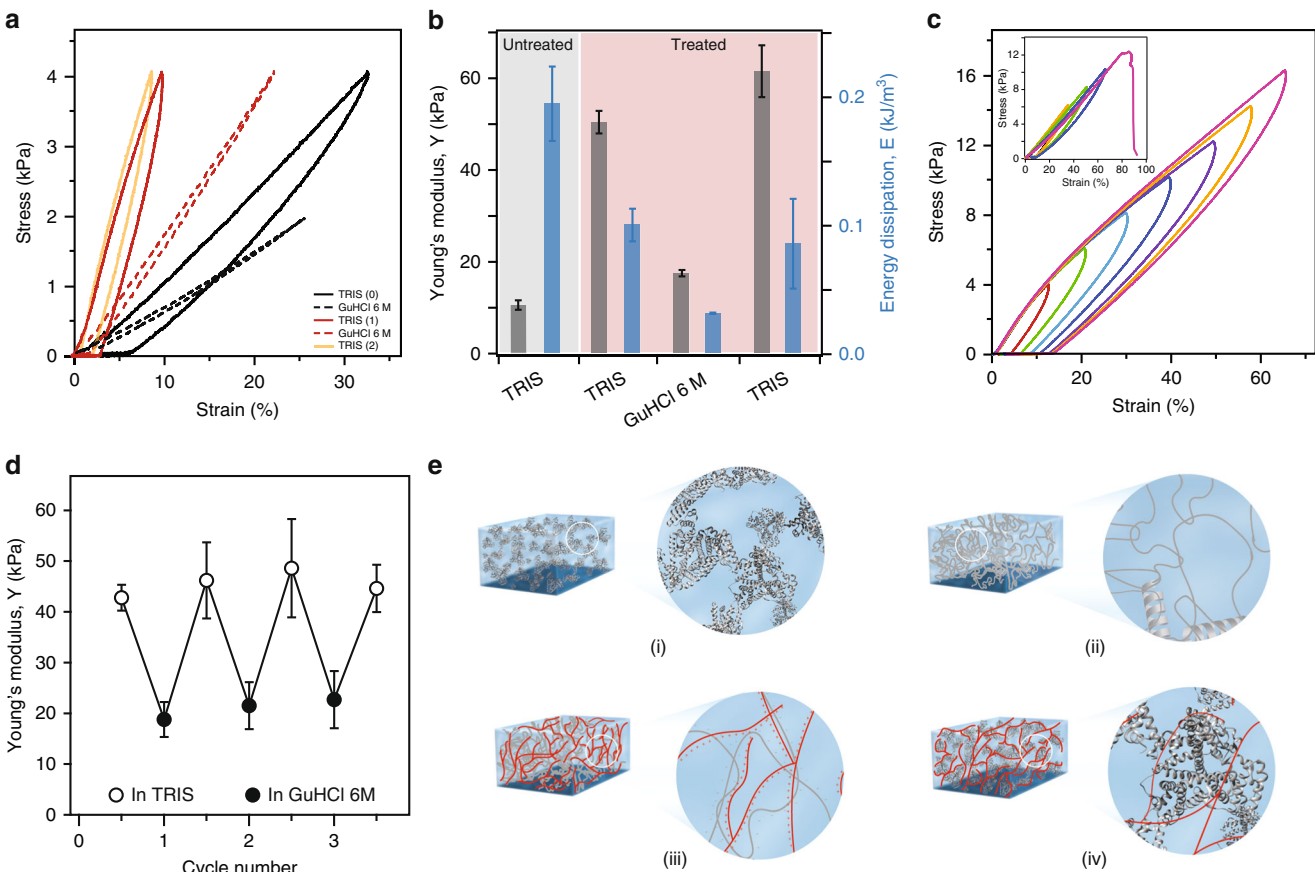

**Fig. 3** PEI stiffens BSA hydrogels without preventing chemical denaturation of protein domains. **a** Stress–strain curves of same BSA-based hydrogel sample in TRIS (black continuous without PEI, red continuous with PEI) and 6 M GuHCl (black dotted without PEI, red dotted with PEI). The gel recovers the initial elastic properties upon immersion from GuHCl back to TRIS (orange trace). This reversible behavior enables shape memory and reversible shape change using chemical (un)folding. **b** Average Young's moduli and energy dissipation of same BSA-based hydrogel sample in TRIS and 6 M GuHCl before and after treatment with 2 mM PEI solution. **c** Stress–strain of PEI-treated BSA-based hydrogel sample showing that it can withstand successive loading–unloading cycles with increasing final stresses while showing a pronounced hysteresis and recovery. Inset: Stress–strain of BSA-based hydrogel sample failed after successive loading–unloading cycles of increasing force. **d** Dynamic change of the Young's modulus of BSA hydrogels treated with 2 mM PEI in response to switching between native (in TRIS) and denaturing (in GuHCl 6 M) buffer conditions. **e** Proposed mechanisms for the strengthening of treated PEI–BSA-based hydrogel: (i) Covalent crosslinking between BSA domains. (ii) Protein (un)folding nanomechanics. (iii) Non-covalent electrostatic crosslinking between unfolded BSA domains and PEI molecules, and (iv) PEI molecules stabilize a BSA domains through electrostatic interactions. All error bars are SD from $n = 3$ independent hydrogel samples.

hysteresis as before the immersion in the GuHCl solution (Fig. 3a, b). Mechanical unfolding of protein domains allows BSA hydrogels, with or without PEI treatment, to maintain their elastic behavior all the way to the breaking force. However, due to its stiffening effect, PEI increases the maximum stress that a BSA hydrogel can sustain (Fig. 3c). In addition, PEI-treated BSA hydrogels can be reversible cycled between native (TRIS) and denaturing (GuHCl 6 M) conditions (Fig. 3d). While PEI-induced stiffening of BSA hydrogels enables the programming in various shapes, this reversible softening and stiffening in the presence and absence of chemical denaturants can produce reversible shape memory.

**Shape memory of BSA–PEI hydrogels.** The PEI–BSA interactions inside the hydrogel matrix provide a vista for constructing moldable and shape-memory biomaterials based on globular proteins. Above, we observed that PEI can strengthen the BSA-based hydrogel sample and exhibits a good recovery (Fig. 3). We are using this phenomenon to program BSA-based hydrogels in various shapes. We demonstrate this approach using different

3D-printed molds (Fig. 4a). A BSA-based hydrogel was immobilized to obtain a spring-like and a W-shape, and programmed by immersing it in a 2 mM PEI solution for 30 min at room temperature. Following a wash step with TRIS buffer, the gel preserved its programmed shape in TRIS buffer (Fig. 4b). To disrupt the programmed shape of the BSA-based hydrogel, we use 6 M GuHCl denaturant solution. The denaturing solution triggers protein unfolding (Fig. 4b, Supplementary Fig. 4) that results in the macroscopic loss of the programmed shape (Fig. 4b). When the hydrogel is moved back into TRIS, the gel regains its PEI-programmed shape after a couple of minutes (Fig. 4b, Supplementary Movie 1).

To further quantify the programming efficiency and cyclicity of the memory loss and regain of our protein hydrogels, we used the standard U-shape approach[31,32]. As shown in Fig. 4c, U-shape gels were successively cycled three times between folding (TRIS) and denaturing conditions (GuHCl 6 M) and the measured bending angle $\theta$ was used to quantify fixity ratio, $R_f$, and shape recovery ratio, $R_r$. The bending angle $\theta$ of the programmed protein hydrogels was $142 \pm 6$ deg in native TRIS buffer and $33 \pm 11$ deg in the denaturing buffer, where the hydrogel loses its shape

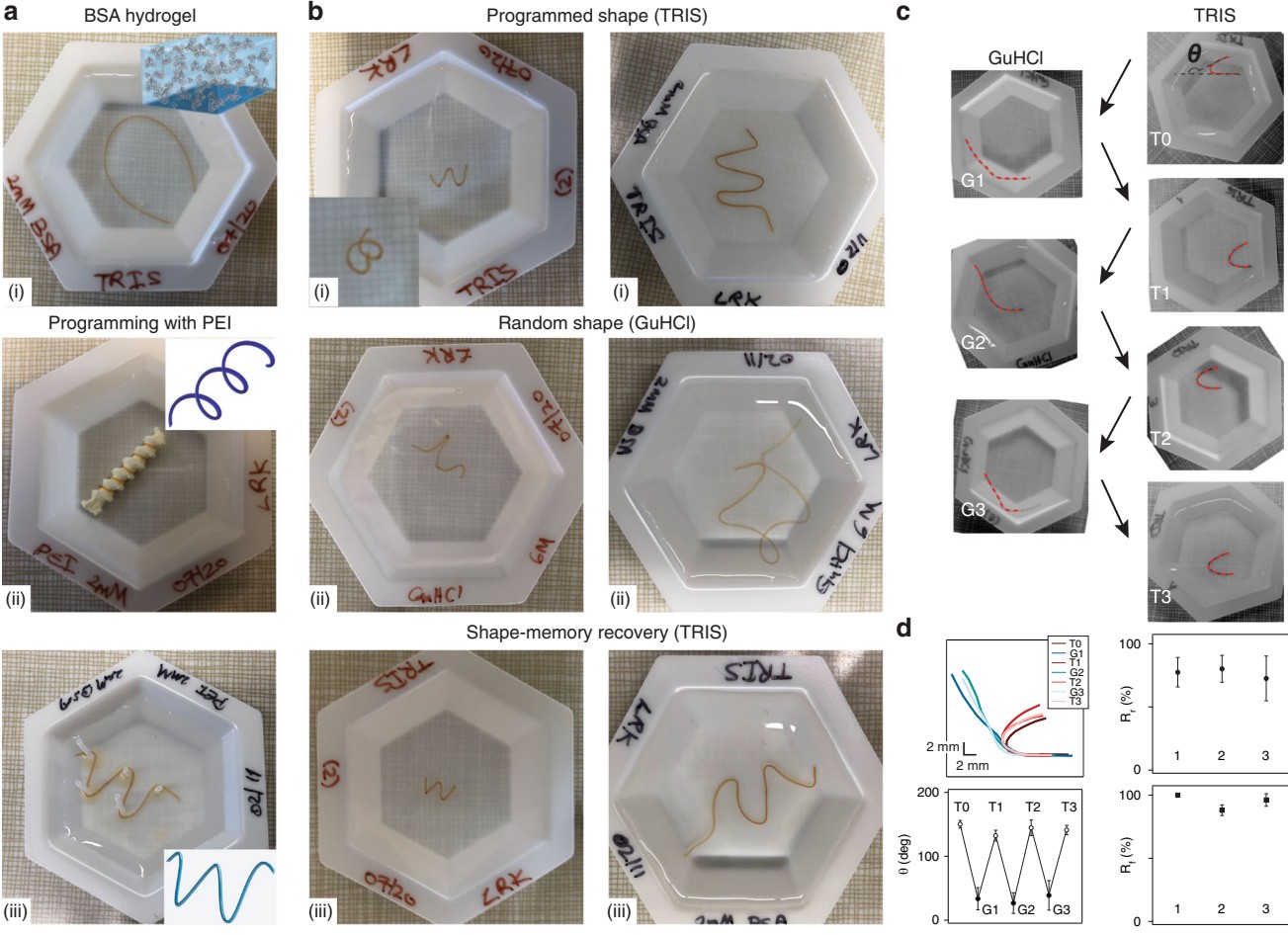

**Fig. 4** Shape-memory recovery process of BSA-2 mM PEI-based hydrogel. **a** (i) BSA-based hydrogel extruded from PTFE tube into TRIS buffer; (ii) BSA hydrogel programmed into a spring-like shape, using a 3D-printed screw-like shape structure, by immersing it in 2 mM PEI solution; (iii) BSA hydrogels programmed into W-like by treating with 2 mM PEI solution. **b** Shape-memory cycle of protein hydrogel programmed as a spring (left) or W-shape (right). Picture of hydrogels in (i) native TRIS buffer; (ii) denaturing GuHCl 6 M solution; (iii) recovered shape in native TRIS buffer. **c** Pictures of a BSA hydrogel programmed in a U-shape, undergoing denaturing/recovering cycles. The gel shape was traced (red) and the bending angle was quantified in both types of buffers. **d** (Top left) Digitized gel shapes from images in **c**; (bottom left): measured bending angle in native TRIS buffer (T) and denaturing GuHCl 6 M buffer (G). The subscript indicates the cycle number. (top right): Recovery ratio $R_r$ as a function of cycle number; (bottom right) the fixity ratio $R_f$ as a function of cycle number. Data were obtained from three different measurements. All error bars are SD from $n = 3$ independent hydrogel samples.

(Fig. 4d). Fixity ratio, $R_f$, which reports on the efficiency of programming by comparing the measured bending angle following denaturation/refolding cycles with the programmed angle, was found to be equal to $95 \pm 6\%$ (Fig. 4d). Shape recovery ratio, $R_r$, reports on the ability of the material to memorize the permanent shape and its capability to cycle between programmed and denatured shapes, was in our case $R_r = 76 \pm 4\%$ (Fig. 4d). These values are comparable to those reported for polymeric materials with thermally induced shape memory[32].

## Discussion

For protein hydrogels made from structured proteins, a unique viscoelastic property comes from the nanoscopic response to force of their constituent domains, which can reversibly unfold and extend 4 to 10 times their initial folded length[14,28]. This unique response to force, combined with their biocompatibility and diverse functional spectrum, place protein hydrogels at the forefront of bioengineering. However, they still have major drawbacks. (i) Protein hydrogels typically show weak mechanical integrity and increasing the number of crosslinking sites can improve their stiffness, but at the expense of a narrower

tunability[14,18,33]. For BSA, the minimum gelation concentration is ~0.7 mM, while the saturation concentration is ~4 mM, which translated into a Young's moduli range between 2.5 and 15 kPa[15]. When treated with PEI, BSA-based hydrogels (2 mM) showed a significant increase in the Young's modulus, up to ~64 kPa (~6-fold increase), and a wide range of stiffness tunability, ranging from 10 to 64 kPa (Fig. 2). (ii) Hydrogels also tend to be weak and brittle, with low dissipation energy and elastic moduli[34]. Several different approaches were proposed for polymer-based hydrogels to circumvent this important flaw, using double[7,35] or triple[36] overlapping networks. Typically, the structural failure of the first network is mitigated by the takeover of the secondary network. Here too, by immersing BSA hydrogels in polyelectrolytes such as PEI, we generate a double-network. The electrostatically driven PEI adsorption to the positive patches of the BSA domains constitutes as a secondary network. The dependency of the Young's modulus and energy dissipation with changing ionic strength reveals a subtle relation between intra- and inter-chain electrostatic interactions and protein folding (Supplementary Fig. 5). Native-BSA hydrogels show a slight decrease in Young's modulus with increasing salt concentration, as more protein domains are folded when moving to native-like

conditions (150 mM ionic strength, Supplementary Fig. 5A, red points)[37]. PEI-treated hydrogels show a decrease in Young's modulus until ~10 mM, followed by a sharp increase in stiffness (Supplementary Fig. 5A, blue points). We speculate that with increasing salt concentration, the PEI-treated BSA gel stiffness decreases due to the screening of electrostatic inter-chain interactions up to 10 mM[38], and increases back due to screening of intra-chain interactions of PEI, leading to its compaction. While increasing salt will also decrease the attractive electrostatic interaction between PEI and the negative patches of BSA, the polyelectrolyte stays bound, probably due to hydrophobic interactions[39,40] and reversible aggregation[41]. This feature plays an important role in the recovery of the programmed shape of the hydrogel, when immersed back from denaturing conditions to TRIS buffer.

Polymer-treated protein hydrogels operate on a different mechanism than other double-network systems, as the primary network can dissipate energy through protein (un)folding nanomechanics, and the secondary polymer network acts to tune the stiffness and reinforce the hydrogel. In our case, PEI adsorption does not only serve as a secondary supportive network but also contributes synergistically to the mechanical stability of the BSA domains inside the hydrogel matrix. As the concentration of PEI increases, the measured hysteresis in the stress–strain curves decays to a constant value at ~0.75 mM of PEI (Fig. 2c). Furthermore, since unfolding and refolding of protein domains is a reversible process, and a large amount of mechanical work can be dissipated during the protein (un) folding transitions, the native-BSA gels do not show plastic permanent deformations until breaking, which occurs at ~11 kPa for PEI-free BSA hydrogels, whereas the PEI treatment increased the breaking force beyond 18 kPa (Fig. 3c). Furthermore, PEI enables reproducible behavior while the final force is increased successively, without impairing the backbone structure of the hydrogel (Fig. 3c). The measured Young's modulus does not vary significantly with buffer between successive cycles. This repeatable response suggests that, if present, partial detachment of PEI has a negligible effect and chemical denaturants do not perturb significantly the interaction between BSA and PEI (Fig. 3d).

The significant increase in stiffness, extensibility, and recovery of BSA hydrogels treated with PEI can now allow for programming these hydrogels into a specific shape. This shape programming can enable these materials to extend their use into different applications such as soft robotics and actuators[12,13], and was accomplished here by mounting the BSA hydrogel on a mold, followed by immersion in PEI solution (Fig. 4). To trigger shape changes, we used the unique response of proteins to chemical denaturants, such as GuHCl 6 M. PEI-incubated BSA hydrogels display a change in stiffness from ~64 kPa in TRIS buffer to ~18 kPa and lose their hysteresis, as shown in stress–strain curves (Fig. 3a). This significant change in stiffness in chemical denaturant leads to loss of the programmed shape, as the BSA domains forming the skeleton are denatured. Importantly, this unfolding process is reversible, as the BSA gels recover both their initial Young's modulus and energy dissipation behavior when immersed back from GuHCl to TRIS (Fig. 3). Macroscopically, this buffer change results in the recovery of the programmed shape during the washout of the GuHCl salts (Supplementary Movie 1). This approach demonstrates that incubation of polyelectrolyte with protein hydrogels does not only increase the attainable stiffness and tunability but also allow hydrogels to operate in a stimuli-responsive manner. Other systems, based on fibrillar proteins such as collagen, use swelling and deswelling to actuate macroscopic movements[42]. Our system is unique, as it is utilizing the reversible unfolding and refolding of protein domains to trigger deformation and recovery of the programed shape.

Finally, the experiments performed here allow us to also understand the synergistic strengthening mechanism. First, covalent crosslinking of BSA molecules at the tyrosine sites produce a network that responds to force in a fully reversible way, in the sampled force range (Fig. 3e (i)). Second, the (un)folding nanomechanics of BSA domains inside the hydrogel matrix allow for large amounts of energy dissipation before physical damage of the network can occur (Fig. 3e (ii)). Third, the non-covalently attached polyelectrolytes can form and break local bonds, allowing the gels to heal any structural damage inside the BSA network caused by the applied stress or strain (Fig. 3e (iii)). Fourth, there is a synergistic effect between PEI strengthening single BSA domains and PEI bridging several protein molecules (Fig. 3b and e (iv)).

In summary, we show the first implementation of a simple method to program the shape of protein hydrogels using polyelectrolytes and to induce a reversible shape change using the unfolding–refolding response via chemical denaturants. This programming is possible due to the stiffening effect that PEI has on BSA hydrogels, which can change the Young's modulus up to sixfold its original value. The unique polymer–protein interaction inside the hydrogel matrix enables shape memory for electrolyte-treated protein hydrogels. While the reversible response is induced here by chemical denaturants, we anticipate that other protein (un)folding specific triggers, such as pH, salt, light, temperature, or external triggers could be introduced in the future. Given the recent developments in designing heteropolymers and the huge library of proteins, it will be possible to generate new smart protein-based hydrogel biomaterials for further use as drug delivery vehicles[43], tissue engineering scaffolds[44], smart actuators as artificial muscles, and soft robotics for delicate bio-applications[12,13], limited largely only by imagination.

## Methods

**Treated BSA-based hydrogel synthesis**. In all, 2 mM BSA-based hydrogel were synthesized inside PTFE tubes (Cole-Parmer) using a light-activated reaction[15]. Then, the hydrogel sample moved from the TRIS buffer (Tris 20 mM, NaCl 150 mM, pH ~7.4) and immersed for 30 min at room temperature (RT) in one of three polymer solution: polyethyleneimine (PEI) MW ~10 kDa, poly-(L)-lysine (PLL) MW ~10 kDa, and polyethylene glycol (PEG) MW ~8 kDa, which were dissolved in TRIS buffer at various concentrations. After treatment process, the hydrogel was moved to TRIS buffer for another 30 min at RT to washout unbounded polymer molecules from the hydrogel sample.

**Mechanical characterization**. The mechanical characterization of the (un)treated BSA-based hydrogel samples were performed by a force-clamp rheometer machine[15,20]. The native-BSA-based hydrogel sample was subjected to a force-ramp protocol with a controlled stress/relaxation rate of 0.01 mN/s at room temperature when the native hydrogel is immersed into TRIS or 6 M GuHCl solution. Thereafter, the same hydrogel sample was treated with one of the three polymers: PEI, PLL, and PEG as described above. Then, the same force-ramp protocol was applied at the hydrogel sample while it is immersed into TRIS or 6 M GuHCl solution at RT. The Young's modulus was calculated from the slope over 2–12% of strain ratio of each stress–strain curve. The energy dissipation was calculated from the hysteresis area that is enclosed in the stress–strain curves.

**Scanning electron microscope**. SEM characterization was performed to study the pore size, shape, and the wall thickness of treated and native-BSA-based hydrogel samples. Native and treated hydrogel samples were prepared, then frozen in liquid nitrogen prior to lyophilization for 24 h. Thereafter, dried samples were broken with forceps to expose the cross-sectional area. Afterwards, the samples were mounted on aluminum stubs using double-side carbon tape. Then, the samples were sputter-coated with 3 nm layer of iridium prior to imaging with SEM (HITACHI S-4800) using 5 keV acceleration voltage. The samples pore-area size and wall thickness were characterized using ImageJ (NIH, USA).

**Water content measurements**. Water content measurements were repeated three times for every sample. Treated BSA-based hydrogels at different concentration of PEI (0.25–2 mM) were synthesized and treated as described in the previous section. Then, the hydrogels were immersed in Tris buffer (20 and 150 mM NaCl, pH ~7.4) at 4 °C for 24 h. Thereafter, the hydrogels were removed from the TRIS solution and excess buffer was removed using filter paper, then weighed to obtain the wet weight ($W_{wet}$) of each sample. Afterwords, the same hydrogels samples were dried using a desiccator for 24 h. Next, the dried samples were weighed, and the dry weight of each sample ($W_{dry}$) was obtained. The swelling ratio (SR) was obtained using the following equation:

$$SR = \frac{W_{wet}}{W_{wet} + W_{dry}} \times 100. \tag{1}$$

**Observation of shape-memory programming and recovery**. The shape-memory programming and recovery was observed as follows: A 2 mM BSA hydrogel was synthesized inside the PTFE tube. The hydrogel sample was then programmed in U-shape, 2D W-like shape or 3D spiral-like shape. Then, the complex was immersed in 2 mM PEI solution for 30 min at RT to program the hydrogel sample. Then, the programmed hydrogel sample was released from the template and immersed into TRIS solution for 30 min at RT to remove any excess PEI molecules from the hydrogel sample. To demonstrate the shape-memory recovery effect, the hydrogel sample was immersed into GuHCl 6 M denaturant for 30 min at RT to lead for a random temporary shape deformation. Afterwards, the sample immersed back in TRIS solution to observe the shape recovery process with time. To quantify the cyclicity of the shape-memory behavior we used the U-shape method[31,32]. In this approach, 2 mM BSA hydrogels were programed in a U-like shape using a mold that produces an expected shape angle of 150.7°. The bending angle θ was measured from images of hydrogels in various solution using ImageJ, as depicted in Fig. 4c top right. The shape fixity ratio $R_f$ and shape recovery ratio $R_r$ are defined as[31]

$$R_f = \frac{\theta_t}{\theta_i} \times 100, \tag{2}$$

$$R_r = \frac{\theta_i - \theta_f}{\theta_i} \times 100, \tag{3}$$

where $\theta_i$ is the actually curled angle, $\theta_t$ is the temporarily fixed angle, and $\theta_f$ is the final angle.

## Data availability

The data that support the findings of this study are available from the corresponding author upon request.

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

## Acknowledgements
We thank Dr. Heather Owen for access to the Electron Microscopy Facility and help in acquiring scanning electron microscopy images. This research was funded by the Greater Milwaukee Foundation (Shaw Award), the University of Wisconsin system (RGI 101X396), and by the National Science Foundation (grant number MCB-1846143).

## Author contributions
L.R.K. performed the measurements. L.R.K. and I.P. designed the research, analyzed the data, discussed the results, and wrote the manuscript.

## Competing interests
The authors declare no competing interests.
