## [Peer Review File · Nature Communications]

Reviewers' comments:

Reviewer #1 (Remarks to the Author):

In this manuscript, Khoury and Popa reported a new method to tune the stiffness of bovine serum albumin (BSA)-based hydrogels by using protein-polymer interactions. They used positively charged polyelectrolytes to successfully increase the stiffness of BSA hydrogels by a ~6 folds. Building upon this result, the authors further proposed to engineer shape memory hydrogels. The results presented in this manuscript are quite interesting, the data analysis was carried out in a careful fashion and the manuscript is well-written. However, the manuscript in its present form lacks any detailed mechanistic analysis, some key results need to be substantiated.

1) The authors used photo cross linking method to create the BSA hydrogel. Since BSA is a monomeric protein, all BSA need to be cross linked into the network to form a hydrogel. However, this part is not discussed. How many tyrosines are involved, what is the cross linking density, and et al.

2) The authors attributed the hysteresis to the unfolding of BSA, however, this result was not supported by any experimental evidence. The authors need to provide convincing evidence to this important claim.

3) The authored observed a six fold increase in modulus of the BSA hydrogel when it interacts with PEI. However, this increase is at the expenses of the extensibility. In hybrid hydrogel, will PEI be detached from the protein hydrogel network during stretching to contribute to the hysteresis? Again, experimental evidence is needed.

4) The shape memory is based on the unfolding/refolding of BSA. First, the shape memory effect needs to be better quantified. At it stands now, there appears to be a tendency to go back to the original shape. However, it is hard to make a claim of shape memory. Second, the mechanism of shape memory does not have any mechanistic investigation. There is no data on the effect of denaturants on the interactions between folded and unfolded BSA with PEI and polylysine. How do denaturants affect the negatively charged patched on BSA? It is not clear how the network will have the memory upon complete unfolding if the interactions between PEI and BSA are altered by the denaturants.

Reviewer #2 (Remarks to the Author):

The manuscript envisioning a new generation of shape-memory material based on polymer protein networks is highly interesting.

Major revisions are needed about the following points:

- Page 4: "synthesized inside semi-transparent tubes": What is the material of the tube? Why are they necessary for crosslinking of tyrosine groups on BSA molecules? These questions should be shortly addressed on the text although there is a reference given.

- Page 4: "due to these phase transitions": It is not clear what kind of "phase transitions" are implied here. If folding/unfolding is being referred by "phase transitions" that should be made more clear to the readers.

- Page 6: Please provide a reason for non-monotonic wall-thickness of the pores with increasing polymer concentration.

- Page 8: The authors attribute the stiffness in BSA-polycation hydrogels to protein folding. Thus, Guanidium HCl, which is a chemical denaturant, is added to break hydrogen bonds that hold together secondary and tertiary structure of proteins. It is also mentioned that electrostatics can be a contributor due to negative charges of BSA and positive charges of polyethyleneimine but contribution of electrostatics is implied to be of minor importance. This conclusion is premature since no experiment was done to test the effect of charge screening. Thus, stress-strain experiments at different ionic strengths should be done to decide on whether its protein "(un)folding mechanics" or elasticity born from protein-polyelectrolyte interactions.

Page 12: "Our system is unique, as it is able to reversibly switch macroscopically between extended and programmed shape in response to the nanoscopic unfolding and refolding of protein domains." This remark is not verified since salt-dependence is not checked. Thus, it is not known for sure the reversible behavior is solely due to unfolding and refolding of protein domains. Please see the comment above.

Page 13: "PEI molecule stabilizes the BSA proteins, acting as a shell around the folded domains": There is no experimental proof or literature reference provided for this remark. If it is a speculation, then it should be noted as such.

Minor revisions are needed about the following points:

- Page 3: " ± 0.6 kPa to ± 3 kPa" : \pm sign is confusing to readers.

- Page 3: "Using this unique interaction between BSA and polymers": "Unique interaction" implies specific interaction between BSA and polymers, which is not what authors probably meant to say. Please paraphrase this sentence to make the meaning more clear.

- Page 4: "It has a negative charged at pH \sim 7.4": Please replace "charged" with "charge".

Page 8: "but the stiffness is higher than that of the native BSA gels in TRIS (\sim 18 kPa vs \sim 10 kPa, Figure 3A and B)." The authors should at least give a hypothesis that explains this observation.

Page 9: The figure legend does not show which sample is before and which sample is after treatment with 2 mM PEI. Also, it is not written explicitly in the experimental section what the third TRIS washing is about.

Page 12: "PEI treatment increased the breaking force beyond 24 kPa, (Figure 3C)." The highest stress on Figure 3C is 18 kPa. Is it supposed to be Figure 2B instead?

Page 15: "The hydrogel sample was programmed in 2D W-like shape or 3D spiral like-shape." Here, the phrase "programmed" creates a misunderstanding that there is computer programming involved when there is only a template that is used. "programmed" should be paraphrased.

Dear Editor,

We would first like to thank you for handling our manuscript, and to thank the reviewers as well for their useful comments and suggestions. Please find below the points raised by the reviewers, marked in *italic*, and our answers, in **bold**. The changes made in the main text are marked in red in the annotated version of the manuscript.

Reviewers' comments:

Reviewer #1 (Remarks to the Author):

In this manuscript, Khoury and Popa reported a new method to tune the stiffness of bovine serum albumin (BSA)-based hydrogels by using protein-polymer interactions. They used positively charged polyelectrolytes to successfully increase the stiffness of BSA hydrogels by a ~6 folds. Building upon this result, the authors further proposed to engineer shape memory hydrogels. The results presented in this manuscript are quite interesting, the data analysis was carried out in a careful fashion and the manuscript is well-written. However, the manuscript in its present form lacks any detailed mechanistic analysis, some key results need to be substantiated. 1) The authors used photo cross linking method to create the BSA hydrogel. Since BSA is a monomeric protein, all BSA need to be cross linked into the network to form a hydrogel. However, this part is not discussed. How many tyrosines are involved, what is the cross linking density, and et al.

This is an excellent point raised by this reviewer, which we addressed in detail in our first manuscript on BSA hydrogels (Khoury et al, Macromolecules 2018), and that we are also now addressing briefly in the main text. As it was previously reported, BSA has a total of 8 tyrosine amino acids that are exposed and can be cross-linked to form hydrogels. While we cannot determine if all these 8 tyrosines are cross-linked in all the BSA domains, we can measure if all protein domains are covalently cross-linked from the presence/absence of plastic deformation effects. Indeed, as we previously reported by us, above 1 mM concentrations, BSA hydrogels show behavior in accord to complete cross-linking and, as such, for this study we chose to use hydrogels made from solutions of 2 mM BSA. We have added in the main text the following sentence to reflect this point:

“BSA has eight exposed tyrosine amino acids that can participate in the cross-linking reaction (da Silva et. al, 2017). These covalently cross-linked hydrogels will form a stable primary network, as at the chosen concentration of 2 mM, all protein domains are cross-linked (Khoury et al, 2018).”

2) The authors attributed the hysteresis to the unfolding of BSA, however, this result was not supported by any experimental evidence. The authors need to provide convincing evidence to this important claim.

A direct correlation between unfolding and hysteresis can be made with the help of Guanidinium hydrochloride (GuHCl) 6M salt, which is a well-known chemical denaturant for proteins. Protein based hydrogels show no hysteresis in the presence of GuHCl 6M, as reported in Figure 3A (dotted curves). A second approach, which we are now reporting as part of supplementary Figure S3, uses 8-Anilino-naphthalene-1-sulfonic acid (ANS). ANS

naturally binds to the secondary structure elements of proteins and has the special property of acting as a FRET donor and emitting blue light (~477 nm) only when bound, while in solution it produces a weak green signal (~520 nm). As shown in the new Figure S3, protein-based hydrogels show the characteristic ANS signal in both native and PEI treated BSA hydrogels, and this signal decreases significantly in the presence of GuHCl 6M, marking the chemical unfolding of protein domains. Apart from the new Figure S3, we have added the following sentence in the main text:

“The disappearance of the hysteresis in stress-strain curves is a benchmark for the lack of tertiary and secondary structure, here induced through chemical denaturation (Figure 3A and B). This interpretation is further supported from the decrease in fluorescence intensity of 8-Anilinonaphthalene-1-sulfonic acid (ANS) (Carlson et al., 2017; Togashi et al., 2010), which reports on the folding of BSA domains (Figure S3).”

and

“Furthermore, treating BSA hydrogels with various amounts of PEI does not affect the folding of BSA domains inside the gel (Figure S3).”

3) The authored observed a six fold increase in modulus of the BSA hydrogel when it interacts with PEI. However, this increase is at the expenses of the extensibility. In hybrid hydrogel, will PEI be detached from the protein hydrogel network during stretching to contribute to the hysteresis? Again, experimental evidence is needed.

Complete detachment of PEI from hydrogel does not seem to occur, as the measured elasticity would change between stress-relaxation traces. This issue was addressed through the measurements in Fig. 3C, where we successively exposed the same hydrogel to increasing stress and the traces overlap over the same force range. To address this issue even further, we have now cycled the same BSA-hydrogel treated with PEI between regular buffer (TRIS) and denaturing buffer (GuHCl 6M). In these new measurements, which are now part of the new Fig 3D, we see that the measured Young’s modulus does not vary significantly with buffer between successive cycles. This repeatable response suggests that, if present, partial detachment has a small/negligible effect. Apart from the new figure, we are addressing this point in the main text, which reads:

“In addition, PEI-treated BSA hydrogels can be reversible cycled between native (TRIS) and denaturing (GuHCl 6M) conditions (Figure 3D). While PEI-induced stiffening of BSA-hydrogels enables the programming in various shapes, this reversible softening and stiffening in the presence and absence of chemical denaturants can produce reversible shape memory.” ... “The measured Young’s modulus does not vary significantly with buffer between successive cycles. This repeatable response suggests that, if present, partial detachment of PEI has a negligible effect (Figure 3D).”

We are also commenting in the discussion part on the BSA/PEI interaction, which reads:

“While increasing salt will also decrease the attractive electrostatic interaction between PEI and the negative patches of BSA, the polyelectrolyte stays bound, probably due to hydrophobic interactions (Seyrek et al., 2003) and reversible aggregation (Curtis et al., 2016). This feature plays an important role in the recovery of the programmed shape of the hydrogel when immersed back from denaturing conditions to TRIS buffer.”

4) The shape memory is based on the unfolding/refolding of BSA. First, the shape memory effect needs to be better quantified. At it stands now, there appears to be a tendency to go back to the original shape. However, it is hard to make a claim of shape memory. Second, the mechanism of shape memory does not have any mechanistic investigation. There is no data on the effect of denaturants on the interactions between folded and unfolded BSA with PEI and polylysine. How do denaturants affect the negatively charged patches on BSA? It is not clear how the network will have the memory upon complete unfolding if the interactions between PEI and BSA are altered by the denaturants.

This excellent point raised by the reviewer is now addressed with a new measurement, where the gel is cycled between memory/'amnesia' conditions, as also explained already in point 3. Following the reviewer's suggestions, we have performed new experiments where we cycled the same hydrogels treated with PEI between regular TRIS (memory) and denaturing GuHCl 6 M (amnesia) buffer. The fact that the measured elasticity does not change between cycles reinforces our shape memory claim. Furthermore, as also explained at point 2, we are addressing the unfolding/refolding interactions using ANS as a reporter. We find that the ANS intensity, which we use as a proxy for domain folding, does not vary with PEI concentration, both in TRIS and denaturing conditions. This lack of variance of intensity with PEI for the GuHCl treated gels suggests that denaturants do not alter significantly the protein-polyelectrolyte interaction and PEI adsorption does not change the number of folded BSA domains

Reviewer #2 (Remarks to the Author):

The manuscript envisioning a new generation of shape-memory material based on polymer protein networks is highly interesting.

Major revisions are needed about the following points:

- Page 4: "synthesized inside semi-transparent tubes": What is the material of the tube? Why are they necessary for crosslinking of tyrosine groups on BSA molecules? These questions should be shortly addressed on the text although there is a reference given.

The tubes used are made from Teflon (Polytetrafluoroethylene/PTFE) and commercially available (Colle-Palmer). The cross-linking reaction used here is initiated by light, and the reaction mechanism is described in ref. (Lv, S. et al, 2010). We are now addressing this point better in the Methods section.

- Page 4: "due to these phase transitions": It is not clear what kind of "phase transitions" are implied here. If folding/unfolding is being referred by "phase transitions" that should be made more clear to the readers.

We are clarifying this point in the main text, by adding "unfolding/refolding phase transitions".

- Page 6: Please provide a reason for non-monotonic wall-thickness of the pores with increasing polymer concentration.

We think the 0.75 mM point is an outlier and have changed this figure to show the error bars as standard deviation (before was S.E.M.). Also, we changed the data analysis of all the electron microscopy images to report S.D. as error bars, as was already the case for the errors from rheometry measurements and updated the figures accordingly. We are addressing this point in the main text, which now states:

“This increase in wall thickness is probably due to an increase in polymer mass inside the hydrogel network, resulting in enhanced cross-linking (Meng et. al, 2017).”

- Page 8: The authors attribute the stiffness in BSA-polycation hydrogels to protein folding. Thus, Guanidium HCl, which is a chemical denaturant, is added to break hydrogen bonds that hold together secondary and tertiary structure of proteins. It is also mentioned that electrostatics can be a contributor due to negative charges of BSA and positive charges of polyethyleneimine but contribution of electrostatics is implied to be of minor importance. This conclusion is premature since no experiment was done to test the effect of charge screening. Thus, stress-strain experiments at different ionic strengths should be done to decide on whether its protein “(un)folding mechanics” or elasticity born from protein-polyelectrolyte interactions.

This is a very important point, and we thank the reviewer for raising it. While electrostatic interactions are very important in low salt concentrations, where the double layer can have tens of nanometers, at 150 mM we did not expect significant effects. We are now verifying this assumption and to address the role of electrostatic interactions we have now performed the measurements suggested by the reviewer. In these measurements, we are reporting the Young’s modulus and energy dissipation of PEI treated BSA and native BSA hydrogels, measured in different ionic strengths (new Figure S5).

The measured Young’s modulus has a non-monotonic behavior, which we attribute to inter and intra-chain interactions (Figure S5A). In the low salt range (0.1 – 10 mM), increasing salt concentration results in a decrease in the stiffness of the hydrogel due to better screening of repulsive inter-chain interactions between PEI molecules (Pericet-Camara et al., 2006), but also in an increase in the number of folded BSA domains (Fullerton et al., 2006). Between 10 – 100 mM, intra-chain interactions of PEI molecules were shown to become the dominant effect (Curtis et al., 2016) and our measured increase in stiffness mirror this result.

The measured energy dissipation, which reports on the absolute number of protein domains unfolding when exposed to a given force, is also reported in the new Figure S5B. Interestingly, in 0.1 mM ionic strength, we measure a similar energy dissipation for BSA and BSA+PEI hydrogels, which suggest that repulsive electrostatic interactions drive most BSA domains in an unfolded state, with intramolecular beta-conformation (Fullerton et al., 2006). From 0.5 mM salt, the total number of mechanically unfolding domains then almost doubles for native BSA hydrogels, while in the PEI treated BSA hydrogels, PEI limits the experienced force-per-molecule.

We changed the main text to reflect this fact, which now reads:

“The electrostatically driven PEI adsorption to the positive patches of the BSA domains constitutes as a secondary network. The dependency of the Young’s modulus and energy dissipation with changing ionic strength reveals a subtle relation between intra and inter-chain electrostatic interactions and protein folding (Figure S5). Native BSA hydrogels show a slight decrease in Young’s modulus with increasing salt concentration, as more protein domains are folded when moving to native-like conditions (150 mM ionic strength, Figure S5A red points) (Fullerton et al., 2006). PEI treated hydrogels show a decrease in Young’s modulus until ~10 mM, followed by a sharp increase in stiffness (Figure S5A blue points). We speculate that with increasing salt concentration, the PEI-treated BSA gel stiffness decreases due to the screening of electrostatic inter-chain interactions up to 10 mM (Pericet-Camara et al., 2006), and increases back due to screening of intra-chain interactions of PEI, leading to its compaction. While increasing salt will also decrease the attractive electrostatic interaction between PEI and the negative patches of BSA, the polyelectrolyte stays bound, probably due to hydrophobic interactions (Seyrek et al., 2003; Gallops et al., 2019) and reversible aggregation (Curtis et al., 2016). This feature plays an important role in the recovery of the hydrogel’s programmed shape when immersed back from denaturing conditions to TRIS buffer.”

Page 12: “Our system is unique, as it is able to reversibly switch macroscopically between extended and programmed shape in response to the nanoscopic unfolding and refolding of protein domains.” This remark is not verified since salt-dependence is not checked. Thus, it is not known for sure the reversible behavior is solely due to unfolding and refolding of protein domains. Please see the comment above.

We do not claim that the loss of programmed shape is ‘solely’ due to unfolding and refolding of proteins. The message that we want to send here is that our implementation is the first to use the chemical unfolding of proteins as a trigger for shape change. We have rephrased this sentence, which now states:

“Our system is unique, as it is utilizing the reversible unfolding and refolding of protein domains to trigger deformation and recovery of the programed shape.”

Page 13: “PEI molecule stabilizes the BSA proteins, acting as a shell around the folded domains”: There is no experimental proof or literature reference provided for this remark. If it is a speculation, then it should be noted as such.

We agree with the reviewer and removed this sentence.

Minor revisions are needed about the following points:

- Page 3: “ ± 0.6 kPa to ± 3 kPa” : \pm sign is confusing to readers.

We have done the suggested change.

- Page 3: “Using this unique interaction between BSA and polymers”: “Unique interaction” implies specific interaction between BSA and polymers, which is not what authors probably meant to say. Please paraphrase this sentence to make the meaning more clear.

We have removed the word “unique” from this sentence.

- Page 4: “It has a negative charged at pH~7.4”: Please replace “charged” with “charge”.

We have done the suggested change.

Page 8: “but the stiffness is higher than that of the native BSA gels in TRIS (~ 18 kPa vs ~ 10 kPa, Figure 3A and B).” The authors should at least give a hypothesis that explains this observation.

We have added a hypothesis, as requested by the reviewer. The sentence now reads: “... , suggesting that the interaction between BSA and PEI remains strong and limits gel extensibility even in harsh conditions. “

Page 9: The figure legend does not show which sample is before and which sample is after treatment with 2 mM PEI. Also, it is not written explicitly in the experimental section what the third TRIS washing is about.

We changed the color of the trace measured when the gel was moved back to TRIS from light red to orange and added numbers in parenthesis, to reflect the measured order. The legend also now reflects which curve is before and which curve is after GuHCl exposure. The second point of the final TRIS wash is actually very important and we are now addressing it better in the manuscript, as explained above. The final TRIS wash is used to measure if the PEI-treated hydrogels recover after denaturing the BSA domains. As the hydrogel displays the same response, this suggests that no PEI is being released and no major structural changes take place during this step. Furthermore, our new experiments where the same hydrogel is cycled between TRIS and GuHCl (new Figure 3D) addresses this point in even more detail.

Page 12: “PEI treatment increased the breaking force beyond 24 kPa, (Figure 3C).” The highest stress on Figure 3C is 18 kPa. Is it supposed to be Figure 2B instead?

We have corrected this typo.

Page 15: “The hydrogel sample was programmed in 2D W-like shape or 3D spiral like-shape.” Here, the phrase “programmed” creates a misunderstanding that there is computer programming involved when there is only a template that is used. “programmed” should be paraphrased.

While we agree with the reviewer that “programming” is seldom referred to “computer programming”, this term was adopted by the material science community to describe the process of acquiring a shape due to the formation of a secondary network and bares no relation to computers. See for example the article of Nojoomi et al. in Nature Communication, 2018, which we are citing as ref. 33.

Apart from the suggested changes, we have also corrected some other typos and rephrased some sentences, without affecting their meaning. All these changes are also marked in red.

We thank again the reviewers for their useful comments and suggestions and we hope that we fully addressed all their points.

Reviewers' comments:

Reviewer #1 (Remarks to the Author):

In this revised manuscript, the authors only partially addressed my comment #1 and 3. My comment #2 and 4 remained largely unaddressed. With regard to the force induced BSA unfolding, the experiments carried out by the authors didn't address the question. 1) there are many potential microscopic origins for the hysteresis observed in tensile testing. The experiment that denatured hydrogel did not show any hysteresis does not prove that the hysteresis comes from BSA unfolding. 2) ANS only indicated that BSA is folded in the hydrogel (which was not a question at all). The stress induced unfolding was not demonstrated in any way. With regard to the shape memory, again, there is no mechanistic investigation or a measure to quantify the degree of shape memory.

Reviewer #2 (Remarks to the Author):

I find the revisions satisfactory for publication.

Dear Editor and Reviewers,

Please find below the points raised, marked in italic, and our answers, in **bold**. The changes made in the main text are marked in red in the annotated version of the manuscript.

Reviewers' comments:

Reviewer #1 (Remarks to the Author):

In this revised manuscript, the authors only partially addressed my comment #1 and 3. My comment #2 and 4 remained largely unaddressed. With regard to the force induced BSA unfolding, the experiments carried out by the authors didn't address the question. 1) there are many potential microscopic origins for the hysteresis observed in tensile testing. The experiment that denatured hydrogel did not show any hysteresis does not prove that the hysteresis comes from BSA unfolding. 2) ANS only indicated that BSA is folded in the hydrogel (which was not a question at all). The stress induced unfolding was not demonstrated in any way.

In our opinion, we have proven that there is a direct relation between domain unfolding/refolding under force and the measured hysteresis. Our conclusion is based on (i) the correlation between the lack of hysteresis in GuHCl 6M measurements, which is a known chemical denaturant; collaborated with (ii) the significant decrease in fluorescence intensity from ANS when a hydrogel is immersed in chemical denaturant from TRIS. However, our results do not exclude the presence of other effects, that may contribute to the measured hysteresis. To accommodate the reviewer's point, we are no longer claiming that the hysteresis is a direct reporter for unfolding/refolding transitions, and have changed the paragraph in question from:

"...the stress-strain curves show important hysteresis, which reports on the energy being dissipated due to these unfolding/refolding phase transitions"

to

"... the stress-strain curves show important hysteresis, which reports on the energy being dissipated **during stress-relaxation cycles".**

Finally, we would like to point out that the origin of the hysteresis measured in stress-strain curves is not relevant for the approach proposed. Here, we use the stress-strain measurements as a way to screen for various polyelectrolytes and determine optimal polyelectrolyte concentrations for treatment of hydrogels. From

these measurements, we take the change in stiffness (Young's modulus, given by the slope) as a reporter. This change in stiffness then becomes the 'secret sauce' that allows us to program protein hydrogels in various shapes, by incubation in polyelectrolytes.

With regard to the shape memory, again, there is no mechanistic investigation or a measure to quantify the degree of shape memory.

In the previous revision, we have quantified the recovery of the mechanical characteristics of protein hydrogels during repeated cycling between TRIS and denaturing buffer. However, we agree with the reviewer that this point can be improved further and we are now providing a direct quantification of the degree of shape memory. In these experiments, a material is programmed into a "U"-shape. Following programming, we cycled the hydrogel between alternating memory/chemical denaturant conditions, and the shape memory was quantified using the change in bending angle. From these new measurements, we have quantified the shape recovery behavior using the shape fixity ratio (R_f) and shape recovery ratio (R_r), which report on the ability of a material to memorize the programmed shape. R_f reports on the efficiency of programming by comparing the expected angle given by the shape of the mold and the obtained angle when the hydrogel is immersed in PEI and following denaturation/refolding cycles. R_r reports on the ability of the material to memorize the permanent shape and its capability to cycle between programmed and denatured shapes (Jiao C. et al, *ACS Appl. Mater. Interfaces* 2018; Chen F., et al, *RSC Adv.*, 2019).

In our experiments, which were repeated three times, a "U"-shaped programmed hydrogel was cycled three times between denaturing and refolding solution conditions. We found a value for R_f of 95 ± 6 % and a value for R_r of 76 ± 4 %. These values are comparable to those obtained for polymeric materials. For comparison, Jiao et al report for shape-memory polymeric materials a R_f of 100% and an R_r between 74 to 89 % (Table S1 in their article).

We have added these new measurements as part of Figure 4 C&D.

Figure 4. Shape-memory recovery process of BSA-2mM PEI based hydrogel. A) (i) BSA-based hydrogel extruded from PTFE tube into TRIS buffer; (ii) BSA hydrogel programmed into a spring-like shape, using a 3D-printed screw-like shape structure, by immersing it in 2 mM PEI solution; (iii) BSA hydrogels programmed into W-like by treating with 2 mM PEI solution. B) Shape-memory cycle of protein hydrogel programmed as a spring (left) or W-shape (right). Picture of hydrogels in (i) native TRIS buffer; (ii) denaturing GuHCl 6 M solution; (iii) recovered shape in native TRIS buffer. C) Pictures of a BSA hydrogel programmed in a “U”-shape, undergoing denaturing/recovering cycles. The gel shape was traced (red) and the bending angle was quantified in both types of buffers. D) (Top left) digitized gel shapes from images in C; (bottom left): measured bending angle in native TRIS buffer (T) and denaturing GuHCl 6M buffer (G). The subscript indicates the cycle number. (top right): Recovery ratio R_r as a function of cycle number; (bottom right) the fixity ratio R_f as a function of cycle number. Data was obtained from three different measurements. Error bars represent S.D.

We have also updated the main text to reflect these new measurements, which now reads:

“To further quantify the programming efficiency and cyclicity of the memory loss and regain of our protein hydrogels, we used the standard “U”-shape approach (Jiao C. et al, 2018; Chen F., et al, 2019). As shown in Figure 4C, “U”-shape gels were successively cycled three times between folding (TRIS) and denaturing conditions (GuHCl 6M) and the measured bending angle θ was used to quantify

fixity ratio, R_f , and shape recovery ratio, R_r . The bending angle θ of the programmed protein hydrogels was 142 ± 6 deg in native TRIS buffer, and 33 ± 11 deg in the denaturing buffer, where the hydrogel loses its shape (Figure 4D). Fixity ratio, R_f , which reports on the efficiency of programming by comparing the measured bending angle following denaturation/refolding cycles with the programmed angle, was found to be equal to 95 ± 6 % (Figure 4D). Shape recovery ratio, R_r , which reports on the ability of the material to memorize the permanent shape and its capability to cycle between programmed and denatured shapes, was $R_r = 76 \pm 4$ % (Figure 4D). These values are comparable to those reported for polymeric materials with thermally induced shape memory (Jiao C. et al, 2018).”

and in Methods section:

To quantify the cyclicity of the shape memory behavior, we used the “U”-shape method (Jiao C. et al, 2018; Chen F., et al, 2019). In this approach, 2mM BSA hydrogels were programed in a U-like shape using a mold that produces an expected shape angle of 150.7 deg. The bending angle θ was measured from images of hydrogels in various solution using ImageJ, as depicted in Figure 4C top right. The shape fixity ratio R_f and shape recovery ratio R_r are defined as (Chen F., et al, 2019):

$$R_f = \frac{\theta_t}{\theta_i} \times 100$$
$$R_r = \frac{\theta_i - \theta_f}{\theta_i} \times 100$$

where θ_i is the actually curled angle, θ_t is the temporarily fixed angle, and θ_f is the final angle.

Reviewer #2 (Remarks to the Author):

I find the revisions satisfactory for publication.

We are pleased that we managed to answer satisfactory to all the points raised by Reviewer 2.

We thank again the two reviewers for their useful input in strengthening our results and the editor for handling our manuscript.

REVIEWERS' COMMENTS:

Reviewer #1 (Remarks to the Author):

the authors have addressed my comments adequately. I am happy to recommend its publication.